# The Diversity of Small Mammals along a Large River Valley Revealed from Pellets of Tawny Owl *Strix aluco*

**DOI:** 10.3390/biology12081118

**Published:** 2023-08-10

**Authors:** Jerzy Romanowski, Dorota Dudek-Godeau, Grzegorz Lesiński

**Affiliations:** 1Institute of Biological Sciences, Cardinal Stefan Wyszyński University, Wóycickiego 1/3, 01-938 Warsaw, Poland; porrkana@wp.pl; 2Institute of Animal Science, Warsaw University of Life Sciences—SGGW, Ciszewskiego 8, 02-787 Warsaw, Poland; grzegorz_lesinski@sggw.edu.pl

**Keywords:** Mammalia, riparian habitats, ecological corridor, landscape complexity, Vistula River, *Strix aluco*, owl pellet analysis, connectivity

## Abstract

**Simple Summary:**

The Vistula River in Central Europe is widely recognized for its high nature value and attracts attention of numerous researchers as an example of a dynamic river flowing in an extensively managed floodplain. In this study of diversity of small mammals along the Vistula valley, we analyzed the diet of tawny owl *Strix aluco*, a common predator considered an efficient collector of rodents and shrews. Altogether 19 species, including 12 rodents, 5 soricomorphs (shrews and relatives), 1 carnivore, and 1 bat species were recorded. High species richness of small mammals can be attributed to a mosaic of agricultural land and the remnants of riparian forest and meadows in the Vistula floodplain. Out of the four most numerous species, two (bank vole *Clethrionomys glareolus* and yellow-necked mouse *Apodemus flavicollis*) are typical forest inhabitants, and two (striped field mouse *A. agrarius* and common vole *Microtus arvalis*) are associated with urban and agricultural land. This study of small mammals indicates the important role of natural large river valleys in the maintenance of local biodiversity.

**Abstract:**

The Vistula River is one of the largest European semi-natural rivers of high ecological value that functions as an ecological corridor. To assess the structure of small mammal communities along the Vistula River, an analysis of the diet of an opportunistic predator, the tawny owl *Strix aluco*, was used. A total of 6355 individuals of 19 species were found, including 5 soricomorph species, 12 rodents, 1 carnivore, and 1 bat species. Tawny owls most frequently caught *Apodemus agrarius*, *Clethrionomys glareolus*, *Apodemus flavicollis*, and *Microtus arvalis*. Rodents dominated small mammal communities (90%), followed by soricomorphs (8%), and the share of Chiroptera was significant (2%). Using Ward’s method in cluster analysis, three clusters of sites with similar mammal communities were identified. The cluster that included 17 study sites with the dominance of agriculture habitats was inhabited by diverse mammal communities with a high number of species. In the cluster composed of three suburban forest sites, mammal communities had the lowest diversity, although the high species richness and the highest shares of the forest species (*A. flavicollis, C. glareolus,* and *Nyctalus noctula*). Mammal communities in the cluster were composed of three urban sites were dominated by *A. agrarius* and *M. arvalis*. The study indicates the high species richness of small mammals in floodplains of the Vistula River and the adjacent areas in central Poland. The floodplain offers suitable habitats for species associated with forests, water bodies, agricultural land, and developed areas. The data collected confirms earlier model predictions about the presence of well-connected local populations of forest mammals along the Vistula River.

## 1. Introduction

Riverine floodplains are among the landscape components of the highest ecological, cultural, and economic importance. They form a system in which the river channel, the river margin, and the river floodplain are interdependent and interact [1]. Riverine floodplains are characterized by high biological diversity [2,3], including small mammals [4]. They are also considered centers of biocomplexity and bioproduction, with higher plant and animal species richness than other landscape units in most regions of the world [5,6]. From a biogeographical point of view, rivers and their floodplains play an important role in maintaining landscape coherence and forming a network providing pathways for the dispersal and migration of species [1,7].

Rivers have been intensively modified for centuries, and as a result, floodplains are among the most degraded ecosystems globally. In Europe and North America, up to 90% of floodplains have been reclaimed and cultivated and have therefore lost their primary function [5]. The riverine landscape has become uniform over large areas, leaving a few semi-natural examples with an undisturbed flood regime and diverse river channels. Fluvial dynamics, including the expansion of surface waters (“flood”), is the driving force that sustains habitat heterogeneity in floodplains, including the formation of islands, sand bars, and riparian forests [8]. These floodplain components are considered the most threatened habitats of high conservation value, important for many specialist species sensitive to ecosystem degradation [9]. Small mammals play an important role in various ecological processes, such as natural succession, competitive interactions, dispersion, and functioning of trophic chains [10,11]. It is known that the diversity and structure of small mammal communities respond to local disturbances and landscape changes [12].

The diversity of communities of small mammals is fostered by the presence of landscapes that form on the border of river valleys and adjacent uplands, where there are variously fragmented forests and anthropogenic (agricultural and urban) environments. Such landscape structural complexity increases the species richness and density of this group of animals [13,14,15]. In areas heavily transformed by man, e.g., as a result of agriculture, the key to maintaining biodiversity is to increase landscape diversity [16], and river valleys are an important element of such diversity. In intensively used agricultural landscapes with extensive arable fields, ecological corridors are important for many small mammals, especially those associated with woodlots [17,18], for example, strip riparian forests in river valleys.

Much work done in landscape ecology demonstrates the need for the conservation of rivers that retain dynamic conditions [19]. The Vistula River, the biggest in the Baltic Sea catchment area (1047 km long), is one of the largest European semi-natural rivers and is recognized as an important ecological landscape corridor in Europe [20]. Diverse riparian vegetation between the dykes and forests located inside the river valley provides a refuge for fauna and flora, including at least 51 mammalian species [21]. An analysis of spatial cohesion of riparian habitats showed that the Vistula valley provides a corridor function not only for birds and fishes but also for mammals. Riparian forests and islands were identified as two key elements that facilitate the dispersion of mammals and provide the potential for gene flow among populations in the valley [22]. While the diversity of many taxa of vertebrate fauna in the Vistula valley is well documented, there is a surprising lack of data on the occurrence of small mammals along the river. Few field studies have been conducted along the river section between Warsaw and Kampinos National Park in central Poland [22,23,24].

In this paper, we analyze the species richness and diversity of small-mammal communities along a large section of the Vistula River valley, based on the analysis of pellets of tawny owls *Strix aluco* Linnaeus, 1758. Our study aims to test the outcomes of habitat modeling of the functioning of the ecological corridor of the Vistula valley carried out in earlier works [21,22]. It was concluded that the high cohesion of habitats facilitates the dispersion of mammals and that riparian forests are key elements to providing functional continuity of the corridor. We hypothesize that: (1) the large river valley provides suitable habitats for mammals associated with forests and agricultural land, and (2) the high spatial cohesion of habitats within the valley is reflected in the structural similarity of small mammal communities along the river.

## 2. Materials and Methods

The study area covers ca. 300 km of the Vistula River valley with adjacent areas from Sandomierz to Czerwińsk, Poland (Figure 1). The study area is characteristic of the middle and lower river course. The river is only partially regulated; there are segments of the river showing a braided pattern with many islands and sand bars [25]. The flow and water level are variable, with short flood periods which occur in early spring (snow melting) and sometimes also in summer, caused by rainstorms. The inundated area is limited by the flood-control dykes formed at a distance of a few hundred meters from the main channel. The area between the dykes is covered by rich semi-natural vegetation: permanent meadows, shrubs, and riparian forests. The floodplain contains a network of diverse aquatic sites with numerous meanders, former channels, and oxbows of various sizes.

We estimated small mammal communities using analysis of pellets of tawny owls *Strix aluco*. Tawny owl pellets were collected at 30 study sites between 2010 and 2016. At 17 of the sites, pellets from under trees used by tawny owls for breeding and roosting were collected year round throughout the study period. At the remaining sites, pellets were collected at the time of an annual inspection of occupied tawny owl nest boxes: at 13 sites between 2010 and 2014, and at 5 sites in 2015 (consisting of the pellets accumulated during the three year period since nest box installation). Standard methods for pellet analysis were used [26,27,28].

Pellet analysis offers an efficient means to sample small mammal communities and is widely used for local and landscape-level assessments of mammalian populations. Heisler et al. [29] have shown that small-mammal community composition was better represented when estimated via pellets compared to estimates from conventional trapping. The tawny owl is an opportunistic forager that hunts in many habitats (closed-canopy forests, open areas, etc.) on a wide range of prey, including arboreal mammals (e.g., Gliridae), which are extremely rarely caught during standard live-trapping. These features make the tawny owl diet a reliable estimator for species diversity in small mammal communities [28,30].

The habitat within a 1000 m radius area around pellet-collection sites was described with CORINE Land Cover (CLC) [31]. Based on the CORINE map, we identified four land-use classes (Table 1). Based on expert knowledge, we changed the land-use classification of two study sites (MLO and BIE) from artificial surfaces to forests and semi-natural areas. The proportion of CLC land-use classes at each study site is presented in Figure 2.

### Small Mammal Community Estimates

We estimated the small mammal relative abundance (RA) as a percentage of the individuals of each taxon in the total number of individuals of all small mammals recorded.

In the comparative analyses of the species richness and diversity, we conducted analyses taking into account individuals identified to species “N^i^”. To avoid bias caused by remarkable differences in sample size and number of identified species per study site, we calculated the indices for unified sample size (the smallest N^i^ = 32) with the rarefaction method [32]. Calculation proceeded with EcoSim version 7.0. [33]. For each study site, we analyzed two indices (both calculated for unified sample size with the rarefaction method): species richness “S^r^” and diversity expressed by the Shannon-Wiener (H^r^) index:H^r^ = −Σ p_i_ ln p_i_
where p_i_ = the number of individuals of one particular species divided by the total number of individuals identified to the species level at the site, i = number of species.

We analyzed the similarity of the species community diversity for 21 sites with samples of minimum N^i^ = 32 (70% of sites included). We used Ward’s method [34], which is a criterion applied in cluster analysis. Based on a calculation of Euclidean distances, we built a hierarchical dendrogram which illustrates clusters of sites with similar mammal communities. To implement Ward’s clustering criterion, the function ‘hclust’ (R Stats Package) with defined method = “ward.D2” was applied [35].

Species habitat selectivity was analyzed with Spearman correlation (Rs) between species RA and the proportion of CLC land use classes. Statistical analyses were performed using R program version R-3.6.1 [36].

## 3. Results

We recorded a total of 6355 individuals of 19 species belonging to four orders: 5719 rodents (Rodentia)—90% of individuals, of which nearly 50% belonged to genus *Apodemus*, 506 soricomorphs (Soricomorpha)—8%, 127 bats (Chiroptera)—2%, and 3 carnivorans (Carnivora)—0.05% (Table 2).

Soricomorphs comprised five species, with *Sorex araneus* Linnaeus, 1758 being the most numerous in this group (379 individuals, RA = 6%). Rodents were represented by twelve species from seven genera and three families. In this group, there were about 20% of individuals identified only to the genus level: *Microtus* spp. (103 individuals) and *Apodemus* spp. (1251 individuals)*. Apodemus agrarius* (Pallas, 1771) (858 individuals) was the most abundant mammal species—13.5%, followed by *Clethrionomys glareolus* (Schreber, 1780) (847 individuals)—13.3%, *Apodemus flavicollis* (Melchior, 1834) (741)—11.7%, and *Microtus arvalis* (Pallas, 1779) (736 individuals)—11.6%. The least represented rodent species (RA < 1%) were *Microtus agrestis* (Linnaeus, 1761)*, Muscardinus avellanarius* (Linnaeus, 1758), and *Microtus subterraneus* (de Sélys-Longchamps, 1836) (Table 2). *Nyctalus noctula* (Schreber, 1774) (79 individuals) dominated among all Chiroptera in the samples (Table 2). The only carnivore registered in the sample was *Mustela nivalis* Linnaeus, 1766 (three individuals).

The number of individuals N^i^ and species N^s^ recorded at study sites varied greatly along the Vistula River floodplain, with the highest number of species N^s^ = 16 at site MLO (Table 3). The standardized species richness S^r^ ranged from 4.7 to 12, and the index of mammalian community diversity H^r^ ranged from 1.21 to 2.16. The highest species richness was recorded at site MLO (S^r^ = 9.9, H^r^ = 2.04), which was dominated by forest (ca. 60%) (Figure 2). Nine out of 11 sites with high mammalian community diversity (H^r^ > 1.8) were located downriver of Warsaw (see Table 3, Figure 1), and all of these sites were characterized by the high number of species recorded (N^s^ from 12 to 16).

Clustering analysis conducted for the 21 study sites revealed that there are similar mammal communities along the stretch of the Vistula River. According to the results of Ward’s method, sites were aggregated into three clusters (Figure 3). Besides the similar mammal communities, the study sites assigned to each of the three clusters had a similar proportion of landscape classes (see Figure 2). Cluster 1 included study sites with a dominance of different agriculture habitats, cluster 2 comprised the sites dominated by forest including two large urban forests (MLO, BIE), and cluster 3 comprised mostly sites with a high proportion of artificial surfaces and agriculture areas including meadows and small inhabited parcels. Clusters 2 and 3 were located in the vicinity of Warsaw in the center of the study area, while the sites aggregated in cluster 1 were distributed along the whole length of the studied area (Figure 1 and Figure 3).

Cluster 1 (including 15 study sites) revealed the most diverse mammal community (H^r^ = 2.26) with the highest number of species N^i^ (Table 4). In this cluster, three dominant species shared a similar proportion: *M. arvalis*, *C. glareolus*, and *Microtus (Alexandromys) oeconomus* (Pallas, 1776) (20.5%, 18.2%, and 15.1%, respectively; Table 4). In cluster 2 (sites MLO, BIE, and JRE) mammal communities had the highest shares of the forest species such as *A. flavicollis* (31.0%), *C. glareolus* (23.7%), and *N. noctula* (2.3%), as compared to clusters 1 and 3 (Table 4). This cluster had the lowest community diversity indices (H^r^ = 2.05) (Table 4). In cluster 3 (sites JUL, BUC, and TAR), the dominant species were *A. agrarius* (25.0%) and *M. arvalis* (16.7%, Table 4).

Landscape analysis revealed no clear pattern of selectivity of commensal rodents: *Mus musculus* Linnaeus, 1758 and *Rattus norvegicus* (Berkenhout, 1769) occurred in all three clusters, and no correlation between their relative abundance and land use was detected. Relative abundance of *A. agrarius* positively correlated with the proportion of urban land use types (R_s_ = 0.30, *p* = 0.034) and negatively with the proportion of agriculture areas (R_s_ = −0.41, *p* = 0.026). *C. glareolus* was identified in nearly all sites, and the positive correlation between relative abundance and proportion of the forest landscape was nearly significant (R_s_ = 0.35, *p* = 0.058). *A. flavicollis* relative abundance was positively correlated with forest and urban areas (R_s_ = 0.40, *p* = 0.026 and R_s_ = 0.41, *p* = 0.025, respectively). On the other hand for this species, the correlation with the agriculture areas was strongly negative (R_s_ = −0.53, *p* = 0.002). The relative abundance of *Apodemus sylvaticus* (Linnaeus, 1758) positively correlated with urban land use (R_s_ = 0.37, *p* = 0.044) but strongly negative with agriculture landscape (R_s_ = −0.44, *p* = 0.014). *Microtus* is a genus of rodents mostly associated with open areas. *M. arvalis* was identified at 27 sites (24% of all individuals were found in the TAR site), and its relative abundance was positively correlated with the proportion of agricultural landscape (R_s_ = 0.40, *p* = 0.026). The relative abundance of *M. oeconomus* was negatively correlated with forest (R_s_ = −0.49, *p* = 0.005) but there was no positive correlation with agricultural landscape.

## 4. Discussion

The data collected show high species richness of small mammals in floodplains of the Vistula River in central Poland. Most of the species (except *Muscardinus avellanarius, Microtus subterraneus*, and *Crocidura leucodon* (Hermann, 1780)) are widely distributed in central Poland, and all species (except *C. leucodon*) were earlier recorded in Kampinos National Park, the largest protected area in the vicinity of the Vistula floodplain [37]. Although the natural vegetation of the middle section of the Vistula River is considerably transformed by humans, the floodplain offers suitable habitats for species associated with forests, water bodies, and agricultural land. Out of the four most numerous species, two (*C. glareolus* and *A. flavicollis*) are typical forest inhabitants, and two (*A. agrarius* and *M. arvalis*) are associated with urban and agricultural land. Interestingly, species associated with forest and open agricultural habitats co-occurred in almost all study sites (except sites BUK and WIS, each with sample size N < 30). Increased landscape diversity in intensively used agroecosystems, such as where large river valleys are present, is a key factor in the formation of rich small-mammal communities [38,39].

Ward’s cluster analysis of the mammal communities revealed a pattern of one big cluster (cluster 1, Figure 3) including 17 sites distributed along the river, and two smaller clusters with three sites each, which were situated in the Warsaw urban agglomeration and its adjacent suburban area. Although these two latter clusters divided sites of cluster 1 into two groups located up- and downriver of Warsaw (see Figure 1), it did not influence the similarity of the mammal communities within cluster 1. This cluster was dominated by *M. arvalis*, which is the most widespread vertebrate species in the European agricultural landscape [40], where it mostly inhabits grassland habitats.

Clusters 2 and 3 were defined in the city and its vicinity. The mammal community within cluster 2 was dominated by typical forest inhabitants: *C. glareolus* and *A. flavicollis* (nearly 50%). The positive relationship of an abundance of these rodents with a proportion of the forest within study sites confirms the established habitat selectivity of both species. *Apodemus flavicollis* is strongly dependent on the forest environment [41], while *C. glareolus* is more common in open areas such as grasslands [42] or agricultural landscapes [43,44]. The large urban wooded areas in Warsaw provided conditions for maintaining the population of *A. flavicollis* which colonized the city agglomeration for 30 years [45,46,47]. The routes of its spread and penetration of urban habitats are not well known, but riparian habitats along the river could play an important role in colonizing city parks. *A. flavicollis* shows features of a strong competitive species concerning other small rodents [10], and in Warsaw, it can reduce the numbers of *A. agrarius*, which has been known as an urban species in Warsaw agglomeration for over four decades [48]. It was dominant in rodent communities of Warsaw suburbia in 1980s [49] and therefore, it was abundant in clusters 2 and 3 containing urban and suburban wooded habitats. In agricultural landscapes far from the city, *A. agrarius* is widely distributed but with a relatively lower contribution in rodent communities. For example, at one site situated 40 km NW of Warsaw, its share was less than 5% among rodents in the 1980s [50], which is reflected in the lower abundance of this species in cluster 1 and preference towards urban over agricultural landscape documented in our study.

Out of the whole sample, *A. sylvaticus* had the biggest share in cluster 3 and very small in other clusters. This can be caused by the fact that urban landscapes expose wood mice to greater fragmentation than arable areas, which leads to population isolation that is not mitigated by the presence of dispersal corridors [51].

The small mammal community of MLO consisted of the highest number of species (16) in the Vistula valley and was characterized by the highest diversity indices. It is worth noting that a high number of *M. subterraneus* (45 individuals) was recorded at this site. It was the only site within the city border where *Mustela nivalis* was recorded (the other two sites were far from Warsaw: OBL and PRZ). The high small mammal biodiversity of this site is probably maintained by the ecological connectivity with Kampinos Forest where rich communities of these animals occur [37].

In addition to the urban areas, the landscape along the Vistula is covered by a mosaic of agricultural land and the remnants of riparian forest and meadows. Farther from the river bed, there are patches of deciduous forest as well as small villages. This mosaic of habitats resulted in the highest species richness as well as the highest biodiversity indices for cluster 1. Within this cluster, *M. avellanarius* was found in three forest sites in the south of the study area. As *M. avellanarius* lives in the understory and well-developed forest edge structure or hedges [52], patches of undisturbed woodlots within the agricultural area can be suitable habitats. Although *M. avellanarius* was previously recorded in a forest in Kampinos National Park between 1980 and 2009 [37], it was absent in all samples collected downriver of Warsaw in this study.

Three rodent species had a relatively high contribution to mammal communities aggregated in cluster 1: *M. arvalis*, *C. glareolus*, and *M. oeconomus*. While the first two species are common inhabitants of the agricultural landscape with woodlots, the third one is a wetland species and occurs in marshes in river valleys, wet sedge meadows, overgrown reedbeds, and alder forests [50,53]. We assume that a higher proportion of *M. oeconomus* in mammal communities from cluster 1 as compared to the other two clusters can be related to better access to riparian meadows. Balčiauskas et al. [54] emphasized that abandoned agricultural areas create suitable habitats for the species, thus promoting its increased abundance and wider distribution. *M. oeconomus* shows a higher tolerance to periodical changes in the presence of water in its habitats [12]. This allows the continuous occurrence of this species in small refugia in floodplains of large river valleys. A study in northeast Poland confirmed the connection of *M. oeconomus* with extensive floodplains [55].

Riparian habitats are key foraging sites for *N. noctula* [56,57,58]. These bats were frequently recorded at many localities in the Vistula River valley and surroundings, both in natural and human-transformed areas. This indicates the important role of river valleys as habitats for this bat species. Other bats also focus their flight activity in sites close to water using them as places where they forage or drink water [4,59,60,61,62,63]. River valleys in agricultural areas are an important element of landscape diversity, which positively affects the food base, and thus the species richness and abundance of bat assemblages [64].

In this study, we recorded all species of terrestrial small mammals that were reported earlier in the vicinity of our study sites except *Arvicola amphibius* (Linnaeus, 1758) [22,23,24,37,46]. *Arvicola amphibius* was absent in our samples, indicating that it is not connected with the habitats of a large river valley at present. Its occurrence in wet forests of the Kampinos National Park adjacent to the northern parts of the study area between 1983 and 2012 [37] suggests that it is associated with more stable marshes, contrary to areas periodically flooded. The Vistula River valley seems to be a negligible ecological corridor for this vole. Moreover, in the last decades, this species decreased in numbers, and many localities have been abandoned in central Poland [65].

The outputs of habitat models [21,22] indicated the potential for large viable populations of mammals in the Vistula valley and high cohesion of riparian forests in the floodplain. The results of our study represent the first large set of field data on small mammals associated with forest habitats that can be related to the model outputs. Two forest rodents and one forest bat species were well represented in the small mammal community and occurred in sites grouped into all clusters (with *C. glareolus* occurring in 100% of sites). Those species were present in sections of the Vistula River lacking large forest areas, where only riparian vegetation was present. This confirms earlier model [21] predictions about the presence of well-connected local populations of forest mammals along the Vistula River.

## 5. Conclusions

A mosaic of agricultural land and the remnants of riparian forest and meadows offers suitable habitats for a rich community of small mammals in the floodplain of a large European river.

Communities of small mammals typical for agricultural habitats dominated in the Vistula River valley and revealed high similarity of species diversity.

Field data on the abundance and diversity of small mammals support the claim that the Vistula River valley provides an important ecological corridor for mammals connected with wooded or wet habitats.

## Figures and Tables

**Figure 1 biology-12-01118-f001:**
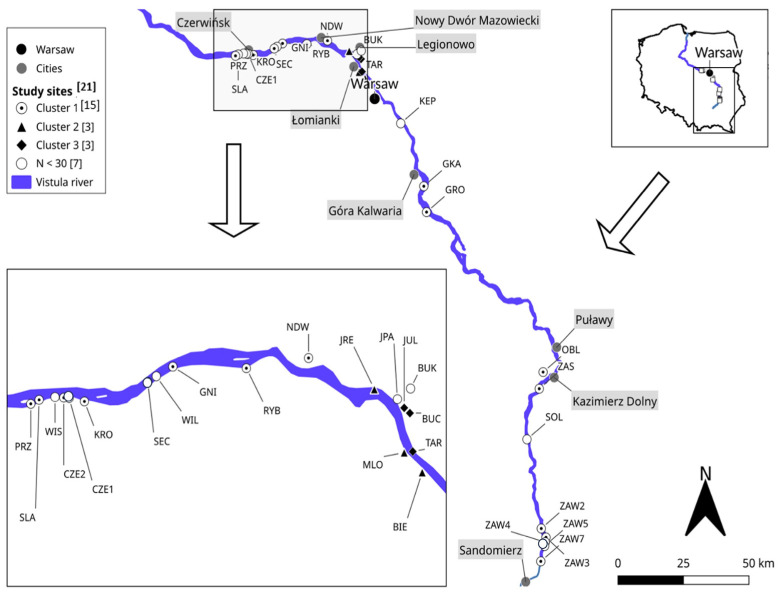
Location of study sites along the Vistula river in Poland. Study sites (with a number of individuals identified to species N^i^ ≥ 32) were analyzed with Ward’s method and assigned to clusters marked with different symbols; the number of study sites in a given cluster is indicated in square brackets.

**Figure 2 biology-12-01118-f002:**
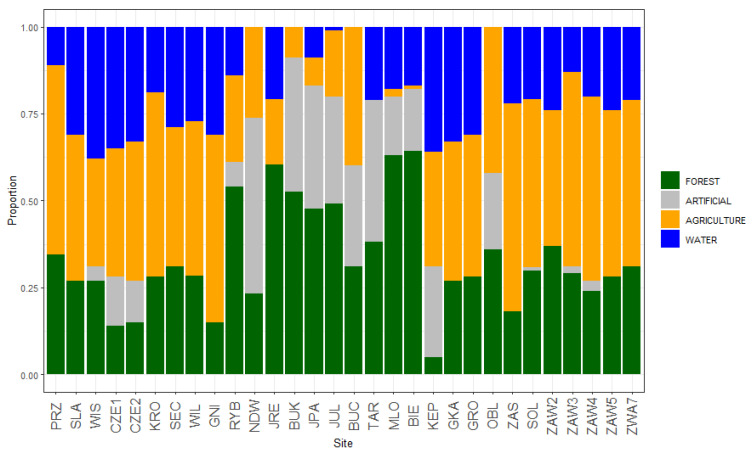
The proportion of CLC classes of land use: forests and semi-natural areas, artificial surfaces, agricultural areas, and water bodies at each of the 30 study sites; study sites in order from up- to downriver.

**Figure 3 biology-12-01118-f003:**
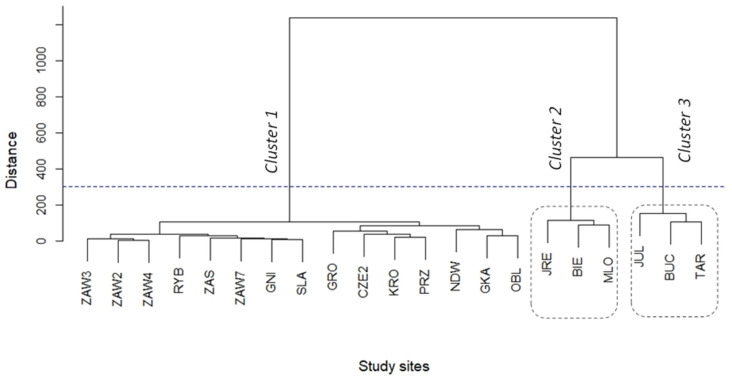
Dendrogram of mammal communities at 21 study sites in the Vistula valley, based on clustering analysis. Only study sites with the number of individuals N^i^ ≥ 32 were included.

**Table 1 biology-12-01118-t001:** CLC classes of land use on studied sites.

Class	Description
Artificial surfaces	Discontinuous urban fabric, industrial or commercial units,
	mineral extraction sites, dump sites, green urban areas, sport and leisure facilities
Agricultural areas	Non-irrigated arable land, fruit trees and berry plantations, grasslands, complex cultivation patterns, land principally occupied by agriculture, with significant areas of natural vegetation
Forests and semi natural areas	Broad-leaved forest, coniferous forest, mixed forest, transitional woodland–shrubland
Water bodies	Water courses, water bodies

**Table 2 biology-12-01118-t002:** Relative abundance (RA) of mammal species identified in owl pellets (N—number of individuals) in the Vistula valley.

Order or Suborder	Family	Species	N	RA (%)	RA (%) Order
Chiroptera	Vespertilionidae	*Nyctalus noctula*	79	1.24	
		Chiroptera spp.	48	0.76	2.00
Soricomorpha	Talpidae	*Talpa europaea*	79	1.24	
	Soricidae	*Sorex araneus*	379	5.96	
		*Sorex minutus*	40	0.63	
		*Neomys fodiens*	6	0.09	
		*Crocidura leucodon*	2	0.03	7.95
Rodentia	Cricetidae	*Clethrionomys glareolus*	847	13.33	
		*Microtus subterraneus*	59	0.93	
		*Microtus* (*Alexandromys*) *oeconomus*	324	5.10	
		*Microtus agrestis*	2	0.03	
		*Microtus arvalis*	736	11.58	
		*Microtus* spp.	103	1.62	
	Muridae	*Mus musculus*	99	1.56	
		*Rattus norvegicus*	75	1.18	
		*Micromys minutus*	422	6.64	
		*Apodemus agrarius*	858	13.50	
		*Apodemus sylvaticus*	190	2.99	
		*Apodemus flavicollis*	741	11.66	
		*Apodemus* spp.	1251	19.69	
	Gliridae	*Muscardinus avellanarius*	12	0.19	90.0
Carnivora	Mustelidae	*Mustela nivalis*	3	0.05	0.05
		Total	6355	100	100

**Table 3 biology-12-01118-t003:** Estimates of small mammal communities from study sites in the Vistula valley. N^i^—number of individuals identified to species, N^s^—number of species recorded, S^r^—species richness calculated for unified sample size (N^i^ = 32) with the rarefaction method, H^r^—species diversity calculated for unified sample size (N^i^ = 32) with the rarefaction method. Study sites are listed in order from up- to downriver.

Study Site	N^i^	N^s^	S^r^	H^r^
PRZ	63	7	6.1	1.44
SLA	32	12	12	2.21
WIS	11	4	-	-
CZE1	22	6	-	-
CZE2	100	13	9.2	1.82
KRO	112	13	8.8	1.87
SEC	24	6	-	-
WIL	10	6	-	-
GNI	32	11	11	2.16
RYB	39	6	5.6	1.30
NDW	162	13	8.6	1.84
JRE	412	12	7.6	1.62
BUK	6	2	-	-
JPA	7	4	-	-
JUL	785	13	8.9	1.98
BUC	854	13	9.3	1.98
TAR	773	13	8.1	1.82
MLO	526	16	9.9	2.04
BIE	406	13	7.5	1.57
KEP	9	5	-	-
GKA	74	10	7.2	1.53
GRO	183	14	8.5	1.90
OBL	110	13	8.3	1.74
ZAS	36	6	5.7	1.27
SOL	8	4	-	-
ZAW2	38	5	4.7	1.21
ZAW3	36	7	6.7	1.46
ZAW4	39	8	7.3	1.44
ZAW5	10	4	-	-
ZAW7	34	10	9.9	2.03

**Table 4 biology-12-01118-t004:** Relative abundance (RA) of mammal species and diversity indices of the Vistula valley mammal communities per cluster analyzed. Only study sites with the number of individuals N^i^ ≥ 32 were included.

Species	Cluster 1		Cluster 2		Cluster 3	
	N^i^	RA (%)	N^i^	RA (%)	N^i^	RA (%)
*Nyctalus noctula*	9	0.8	31	2.3	39	1.6
*Talpa europaea*	15	1.4	27	2.0	35	1.5
*Sorex araneus*	99	9.1	80	6.0	196	8.1
*Sorex minutus*	15	1.4	11	0.8	13	0.5
*Neomys fodiens*	5	0.5	1	0.1	0	0
*Crocidura leucodon*	2	0.2	0	0	0	0
*Clethrionomys glareolus*	198	18.2	318	23.7	306	12.7
*Microtus subterraneus*	7	0.6	51	3.8	1	0.0
*Microtus* (*Alexandromys*) *oeconomus*	165	15.1	47	3.5	98	4.1
*Microtus agrestis*	2	0.2	0	0	0	0
*Microtus arvalis*	223	20.5	94	7.0	403	16.7
*Mus musculus*	41	3.8	10	0.7	44	1.8
*Rattus norvegicus*	15	1.4	24	1.8	34	1.4
*Micromys minutus*	119	10.9	35	2.6	254	10.5
*Apodemus agrarius*	69	6.3	178	13.2	603	25.0
*Apodemus sylvaticus*	5	0.5	19	1.4	166	6.9
*Apodemus flavicollis*	87	8.0	417	31.0	220	9.1
*Muscardinus avellanarius*	12	1.1	0	0	0	0
*Mustela nivalis*	2	0.2	1	0.1	0	0
Number of individuals N^i^	1090	100.0	1344	100.0	2412	100.0
Number of species N^s^	19		16		14	
Rarefaction, N^i^ = 1090						
Species richness S^r^	19		15.7		13.5	
Diversity H^r^	2.26		2.05		2.17	

## Data Availability

Data are contained within the article.

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
