# Peer review of "The Diversity of Small Mammals along a Large River Valley Revealed from Pellets of Tawny Owl Strix aluco"

_biology, 2023, doi:10.3390/biology12081118_

Round 1

Reviewer 1 Report

1. It is advisable to give the author and the year of the original description when writing the Latin name of the species for the first time.

2. Microtus oeconomus - this species belongs to the genus Alexandromys (Kryštufek, Shenbrot, 2022)

3. In the discussion, it is desirable to compare the data obtained based on the analysis of Strix aluco pellets with the literature data on the faunal analysis of the river floodplain obtained as a result of direct captures.

Author Response

The authors acknowledge the careful and valuable comments of all the Reviewers. The authors accepted all corrections and most of the suggestions. Below is the detailed list of all revisions undertaken.

Response to the reviewers' comments

Reviewer 1

  1. It is advisable to give the author and the year of the original description when writing the Latin name of the species for the first time. – we added the author and the year of the original description of all the species mentioned for the first time in the full text.

  1. Microtus oeconomus - this species belongs to the genus Alexandromys (Kryštufek, Shenbrot, 2022) – we added the suggested genus name in parenthesis Microtus (Alexandromys) oeconomus (Pallas, 1776)

  1. In the discussion, it is desirable to compare the data obtained based on the analysis of Strix aluco pellets with the literature data on the faunal analysis of the river floodplain obtained as a result of direct captures. – according to this remark we added the dating of earlier studies in the following sentences:

It was dominant in rodent communities of Warsaw suburbia in 1980-ies [49]…

For example, at one site situated 40 km NW of Warsaw, its share was less than 5% within rodents in 1980-ies [50],…

Although M. avellanarius was earlier recorded in a forest of Kampinos National Park between 1980 - 2009 [37],…

Reviewer 2 Report

I think this manuscript is well done and structured. It represents an excellent source of data for the Vistula valley, both in relation to micro-mammal communities, including riparian bat species, and, of course, regarding the food spectrum of the tawny owl.

Insert between the keywords, as before, 'Strix aluco'

Perhaps it would be more appropriate to transfer the manuscript and its forthcoming publication to the journal 'Land', as it seems to me that more than biology it deals with the ecology of mammal communities, and the ecology of the owl, in relation to the landscape.

Author Response

The authors acknowledge the careful and valuable comments of all the Reviewers. The authors accepted all corrections and most of the suggestions. Below is the detailed list of all revisions undertaken.

Response to the reviewers' comments

Insert between the keywords, as before, 'Strix aluco' – inserted

Perhaps it would be more appropriate to transfer the manuscript and its forthcoming publication to the journal 'Land', as it seems to me that more than biology it deals with the ecology of mammal communities, and the ecology of the owl, in relation to the landscape. – we intend to publish the manuscript in „biology”, as accepted by the remaining reviewers

Reviewer 3 Report

The manuscript by Jerzi Romanowski et al. deals with the diversity of small mammals along the Vistula river valley, using the Tawny owl pellets (Strix aluco). The topic fits well to the focus of the Biology journal, and specifically for its Conservation Biology and Biodiversity section.

In the intensively used landscape of Central Europe, river flows play a key role for biodiversity. It is a mosaic of biotopes, which is dynamically formed in place and time, mainly by water. Periodic inundation and fluctuations in the level and intensity of the water current set the limits even for most human activities. Thanks to this, the floodplain and the river's surroundings enable the long-term existence of populations of a wide range of organisms, incl. many species of small mammals. The manuscript summarizes and compares the data obtained during 7 years on a 300 km section of the middle course of the Vistula River. It is evident the area around the river serves as an important biocorridor and refugium for small mammals as well.

I am not a native English speaker but the text is understandable to me except for the parts mentioned below.

I have some suggestions that I believe could improve the resulting article. They are listed chronologically, resp. in the order in which they appear in the text.

L49-50 - “in lateral and longitudinal directions” seems redundant to me

L52 – I would be more careful in the formulation and omit the word "any"

L65-66 – please try to be more specific (what population processes?)

L77 – please remove the citations

L110 – please simplify; do you mean to say that the dykes are within a few hundred meters of the stream?

L120-123 – This section needs to be expanded. Were pellets from all sites available for each year? How many pellets were available from each location in a particular year? This is important because many species of small mammals are subject to multiannual changes in abundance (population cycles); the absence of records from some years could therefore have significantly affected the results. Please elaborate in more detail.

L142 and in several other places: I think it would be more accurate to talk about the study sites “down the river” or “up the river”; the flow of the Vistula does not have a north-south orientation in the entire area.

L215 – do you mean Tab. 4?

L234 – do you mean TAR site?

L320 – are there any other species whose occurrence was not detected and yet could have been expected? Crocidura spp., Neomys spp. etc.?

I perceive the expansion of the methodological part (L120-123) as a necessary condition for the publication of the manuscript.

Author Response

The authors acknowledge the careful and valuable comments of all the Reviewers. The authors accepted all corrections and most of the suggestions. Below is the detailed list of all revisions undertaken.

Response to the reviewers' comments

I have some suggestions that I believe could improve the resulting article. They are listed chronologically, resp. in the order in which they appear in the text.

L49-50 - “in lateral and longitudinal directions” seems redundant to me – we removed “in lateral and longitudinal directions”

L52 – I would be more careful in the formulation and omit the word "any" – we removed “any”

L65-66 – please try to be more specific (what population processes?): we changed the sentence to: “Small mammals play an important role in various ecological processes, such as natural succesion, competitive interactions, dispersion, and functioning of trophic chains

L77 – please remove the citations – we removed the citation of author’s names

L110 – please simplify; do you mean to say that the dykes are within a few hundred meters of the stream? – we changed the sentence to: “The inundated area is limited by the flood-control dykes formed at a distance of a few hundred meters from the main channel”

L120-123 – This section needs to be expanded. Were pellets from all sites available for each year? How many pellets were available from each location in a particular year? This is important because many species of small mammals are subject to multiannual changes in abundance (population cycles); the absence of records from some years could therefore have significantly affected the results. Please elaborate in more detail. – we added more details on the availability of the pellets in the following sentence: Tawny owls pellets were collected at 30 study sites between 2010 and 2016. At 17 of the sites, pellets from under the trees used by Tawny owls for breeding and roosting, were collected all year round throughout the study period . At the remaining sites, pellets were collected at the time of an annual inspection of occupied by Tawny owls nest boxes: at 13 sites between 2010-2014, and at 5 sites in 2015 (these consisting of the pellets accumulated during the three years period since nest boxes installation).

Please note that „the number of pellets” is not a practical indication of sample size as both pellets and fragments of pellets are available for collection. It is a good practice to analyze both complete pellets and all the available fragments. As a result, the number of identified prey individuals is the best indicator of sample size and we provided these numbers in Table 3. The smallest samples were not included in the further analysis, and we applied rerefaction methods to control for sample size.

L142 and in several other places: I think it would be more accurate to talk about the study sites “down the river” or “up the river”; the flow of the Vistula does not have a north-south orientation in the entire area. We changed the sentences according to this remark

L215 – do you mean Tab. 4? – Yes, we corrected it to Tab.4

L234 – do you mean TAR site? Yes, we corrected it to TAR site

L320 – are there any other species whose occurrence was not detected and yet could have been expected? Crocidura spp., Neomys spp. etc.? No, we discussed the absence of the Arvicola amphibius in the section of the discusion.

I perceive the expansion of the methodological part (L120-123) as a necessary condition for the publication of the manuscript. – we expanded this part.

Reviewer 4 Report

This is an interesting, well-presented study in a dynamic natural/seminatural ecosystem. The methodology (collection of owl pellets) is fully appropriate and the authors analyse the contents of the pellets using standard methods. The sample size is very large (6355 individual prey from 21 sites) allowing for robust analysis and the study area is extensive (300 km of floodplain) so the results have significant validity. The analytical methods are also appropriate (e.g. Shannon-Wiener index, cluster analysis). The Discussion section covers all the main findings clearly and thoroughly and the three Conclusions are fully justified. 

I have no further comments.

The language quality is very good. Minor points: It is conventional to use 'the' before the name of a river. The authors do this in many places in the text and I recommend they add 'the' before Vistula or river Vistula where it is missing, e.g. in lines 12, 19, 24, 104, 243, 323.

Abstract, line 36, please add 'were' between 'cluster' and 'composed'.

I also recommend inserting 'of' between 'NW' and 'Warsaw' in Line 277.

Author Response

The authors acknowledge the careful and valuable comments of all the Reviewers. The authors accepted all corrections and most of the suggestions. Below is the detailed list of all revisions undertaken.

Response to the reviewers' comments

 Minor points:

It is conventional to use 'the' before the name of a river. The authors do this in many places in the text and I recommend they add 'the' before Vistula or river Vistula where it is missing, e.g. in lines 12, 19, 24, 104, 243, 323. - We added the missing „the” in the text.

Abstract, line 36, please add 'were' between 'cluster' and 'composed'. – we added „were”

I also recommend inserting 'of' between 'NW' and 'Warsaw' in Line 277. – we added „of”

Round 2

Reviewer 3 Report

The authors followed most of the reviewers' suggestions and modified the manuscript accordingly. For the remaining comments, they provided detailed answers and explanations of their action.

Nevertheless, I will return to two points where we may have misunderstood each other:

- L145 and 199: north/south vs. down/up the river. In the labels of Fig. 2 and Tab. 3 you write about north-south orientation, in the text about localities down-up the river (L193, 264, 308). It would certainly be appropriate to unify this. I consider the down/up the river version to be more appropriate.

- L328 – I asked whether small mammal species that were not found in the pellets in this study had previously been detected in the vicinity of some of the monitored sites (besides the named Arvicola amphibious, e.g., Crocidura suaveolens, Neomys milleri). Even if there is no other such species, it would be appropriate to mention it in the text.

I am not a native English speaker; however, I am not sure of the linguistic correctness and clarity of some parts of the manuscript (e.g., L12: Central European the Vistula river... and L281: 1980-ies). Therefore, I recommend checking the final version of the text by a native speaker.

These changes fit the criteria of a minor revision.

Author Response

The authors acknowledge the careful and valuable comments of the Reviewer. Below is the detailed list of all revisions undertaken.

Response to the reviewers' comments

Reviewer 3

The authors followed most of the reviewers' suggestions and modified the manuscript accordingly. For the remaining comments, they provided detailed answers and explanations of their action.

Nevertheless, I will return to two points where we may have misunderstood each other:

- L145 and 199: north/south vs. down/up the river. In the labels of Fig. 2 and Tab. 3 you write about north-south orientation, in the text about localities down-up the river (L193, 264, 308). It would certainly be appropriate to unify this. I consider the down/up the river version to be more appropriate. – we corrected the labels of Fig. 2 and Table 3 according to this suggestion: …  study sites in order up-down the river

- L328 – I asked whether small mammal species that were not found in the pellets in this study had previously been detected in the vicinity of some of the monitored sites (besides the named Arvicola amphibious, e.g., Crocidura suaveolens, Neomys milleri). Even if there is no other such species, it would be appropriate to mention it in the text. – according to this suggestion we added the sentence: In this study, we recorded all species of terrestrial small mammals that were earlier reported in the vicinity of our study sites except Arvicola amphibious [22-24, 37, 46].

- I am not a native English speaker; however, I am not sure of the linguistic correctness and clarity of some parts of the manuscript (e.g., L12: Central European the Vistula river... and L281: 1980-ies). Therefore, I recommend checking the final version of the text by a native speaker. – according to this suggestion the native speaker advised us to correct the indicated lines to “The Vistula river in Central Europe”, and the form 1980-ies to “1980s”.